# Efficient-3Dim: Learning a Generalizable Single-image Novel-view Synthesizer in One Day

**Yifan Jiang[1,2]\*, Hao Tang[1], Jen-Hao Rick Chang[1], Liangchen Song[1],**
**Zhangyang Wang[2], Liangliang Cao[1]**
[1]Apple, [2]University of Texas at Austin
`{yifanjiang97, atlaswang}@utexas.edu`
`{hao_tang,jenhao_chang,liangchen_song,llcao}@apple.com`

## Abstract

The task of novel view synthesis aims to generate unseen perspectives of an object or scene from a limited set of input images. Nevertheless, synthesizing novel views from a *single* image remains a significant challenge. Previous approaches tackle this problem by adopting mesh prediction, multi-plane image construction, or more advanced techniques such as neural radiance fields. Recently, a pretrained diffusion model that is specifically designed for 2D image synthesis has demonstrated its capability in producing photorealistic novel views, if sufficiently optimized with a 3D finetuning task. Despite greatly improved fidelity and generalizability, training such a powerful diffusion model requires a vast volume of training data and model parameters, resulting in a notoriously long time and high computational costs. To tackle this issue, we propose **Efficient-3DiM**, a highly efficient yet effective framework to learn a single-image novel-view synthesizer. Motivated by our in-depth analysis of the diffusion model inference process, we propose several pragmatic strategies to reduce training overhead to a manageable scale, including a crafted timestep sampling strategy, a superior 3D feature extractor, and an enhanced training scheme. When combined, our framework can reduce the total training time from 10 days to less than **1 day**, significantly accelerating the training process on the same computational platform (an instance with 8 Nvidia A100 GPUs). Comprehensive experiments are conducted to demonstrate the efficiency and generalizability of our proposed method.

## 1 Introduction

Creating an immersive experience with free-viewpoint interaction, particularly from just one image, has always been a captivating and important area of focus. To achieve realistic interactive visuals, the process can start by constructing a precise 3D model, such as a mesh or point cloud, from the given image, followed by rendering the scene from a new perspective. However, this approach frequently misses out on view-dependent effects and may appear unnatural. Lately, neural rendering techniques, such as neural radiance field (NeRF) (Mildenhall et al., 2020) and scene representation networks (SRN) (Sitzmann et al., 2019), have prevailed over all other approaches and become the crucial step in reaching interactivity and realism. Instead of constructing an explicit representation, these approaches learn an implicit representation by modeling a 3D scene as a continuous function.

Although NeRF-like models produce compelling details, they are impeded by a stringent requirement—the vanilla NeRF approach requires hundreds of posed images for training and generates photorealistic images on both rendered objects and real-world scenes. Several follow-up works reduce NeRF's training views to as few as three views, by using geometry regularization (Niemeyer et al., 2022), Structure-from-Motion (SfM) initialization (Deng et al., 2022), or adaptive positional encoding (Yang et al., 2023). However, these approaches cannot be easily extended to support a single input view. Yu et al. (2021) first propose to train a NeRF that can render images from novel

---

*This work was performed while Yifan Jiang interned at Apple.

viewpoints using a single input, yet it only works on simulated scenes with simple geometry (Chang et al., 2015). Xu et al. (2022) further extends it to in-the-wild scenes and enables novel view rendering on arbitrary objects. However, it only renders views from a small range of angles.

Recently, Watson et al. (2022) proposed the 3DiM pipeline that treats the single-image novel view synthesis as a conditional generative process. By training a diffusion model conditioned on an input image together with the target camera pose relative to the input image, they can synthesize novel views of simple synthetic objects in high fidelity. Liu et al. (2023b) further extend the method to real-world objects by finetuning a text-to-image model to learn geometry priors using 3D simulated objects. Their proposed zero 1-to-3 method can perform zero-shot novel view synthesis in an under-constrained setting, synthesizing view for any in-the-wild image from any angle.

Despite their capability to generate high-fidelity images, training diffusion models for single-image novel view synthesis requires a notoriously long time and prohibitive computational costs. For example, the 3DiM model (Watson et al., 2022) contains 471M parameters; the zero 1-to-3 method (Liu et al., 2023b) uses a 1.2B-parameter Stable Diffusion model (Rombach et al., 2022) and 800k simulated objects from the Objaverse dataset (Deitke et al., 2023). Training the model requires 10 days on a single instance with eight Nvidia A100 GPUs. In comparison, training typical image classifiers (He et al., 2016; Xie et al., 2017; Huang et al., 2017) on the ImageNet dataset (Deng et al., 2009) only takes about 1 day on a similar platform. A straightforward way to shorten the training procedure of diffusion models is to adopt a larger batch size, however, it further increases the required computational resources. Given the importance of diffusion modes and their applications in 3D vision, an enhanced training approach would greatly aid researchers in speeding up the development process, thus propelling the advancement of this direction.

The major goal of this work is to trim down the training time without spending more costs on the total training resources (e.g., taking large-batch via a distributed system). We present **Efficient-3DiM**, a refined solution developed to enhance the training efficiency of diffusion models for single-image novel view synthesis. This paper delves into the details and design choices of the framework. Our approach builds upon three core strategies: a revised timestep sampling method, the integration of a self-supervised vision transformer, and an enhanced training paradigm. Together, these elements contribute to a noticeable improvement in training efficiency. With the proposed enhancements, our model requires only a **single day** to train. Compared to the original zero 1-to-3, we achieve a **14x** reduction in training time in the same computation environment. The significantly reduced training time enables future research to be rapidly iterated and innovated.

Our major contributions are encapsulated as follows:

- We introduce a novel sampling strategy for diffusion timesteps, deviating from the standard uniform sampling, and offering optimized training.

- Our framework integrates a self-supervised Vision Transformer, replacing the conventional CLIP encoder, to better concatenate high-level 3D features.

- Comprehensive experiments have been conducted to demonstrate the effectiveness of our method on various benchmarks. As a result, Efficient-3DiM reduces the training time from 10 days to less than 1 day, greatly minimizing the training cost to a manageable scale.

## 2 RELATED WORK

**Diffusion Model.** Recent breakthroughs in diffusion models (Sohl-Dickstein et al., 2015; Song & Ermon, 2019; Ho et al., 2020) have displayed impressive outcomes in generative tasks. These models, as advanced generative tools, produce captivating samples using a sequential denoising method. hey introduce a forward mechanism that infuses noise into data sets and then reverses this mechanism to restore the initial data. Following this, numerous research efforts (Karras et al., 2022; Saharia et al., 2022; Dhariwal & Nichol, 2021; Nichol et al., 2021; Rombach et al., 2022; Ramesh et al., 2022; Chen, 2023) have been directed toward enhancing the scalability of diffusion models and speeding up the sampling process for improved efficiency. Notably, the LDM model (Rombach et al., 2022) (also known as Stable Diffusion) stands out, minimizing computational demands by executing the diffusion method on a lower-resolution latent space, making the training of a text-to-image model scalable to the billion-scale web data. Several other investigations have also adapted

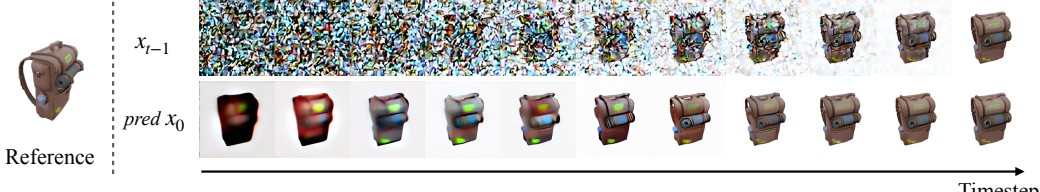

Figure 1: Inference visualization of a typical diffusion-based novel-view synthesizer (Liu et al., 2023b). We run a total of 200 steps following Equation 3 and show the intermediate inference progress of a "backpack" under different timesteps.

the diffusion methodology to diverse fields, such as music creation (Huang et al., 2023), reinforcement learning (Wang et al., 2022), language generation (Li et al., 2022), text-to-3D object (Poole et al., 2022), novel view synthesis (Watson et al., 2022; Xu et al., 2023), video generation (Blattmann et al., 2023), and more. In addition to generation, newer research (Meng et al., 2021; Zhang & Agrawala, 2023; Hertz et al., 2022; Mokady et al., 2022; Brooks et al., 2022; Parmar et al., 2023; Goel et al., 2023; Khachatryan et al., 2023) also extends the capabilities of diffusion models to tasks related to image and video editing.

**Efficient Training for Generative Models.** Existing research efforts have been undertaken to explore various aspects including data efficiency, model efficiency, training-cost efficiency, and inference efficiency. To address the mode collapse issue under the setting of limited training data, Zhao et al. (2020) propose a differentiable augmentation strategy that only takes 10% training data to train a Generative Adversarial Network (GAN) Goodfellow et al. (2014) to achieve similar performance. Nevertheless, it is unclear whether this method achieves a shorter training time. In addition, since low-resolution images are typically more available than their high-resolution counterparts, Chai et al. (2022) propose an any-resolution framework. This framework merges data of varied resolutions to facilitate GAN training, lessening reliance on high-resolution input. However, its impact on training costs and its adaptability for diffusion models have not been thoroughly examined. Another line of research addresses the memory issue during the training stage. Lin et al. (2019) was the pioneer in integrating the patch-wise training approach into GAN frameworks. However, due to their discriminator having to amalgamate multiple generated patches, its memory-saving efficiency is not significant. Building on this, Lee et al. (2022) applied the patch-wise training to the implicit representation framework Skorokhodov et al. (2021). While this method does conserve GPU memory, it compromises the quality of the generated samples. Several researchers have sought to mitigate the substantial training and inference expenses tied to diffusion models. Rombach et al. (2022), for instance, opted to apply diffusion in a latent space, bypassing the pixel space, which effectively cut down both training and inference costs. Other studies, like those of (Lu et al., 2022a;b; Song et al., 2020a; Bao et al., 2022b;a), delved into rapid sampling techniques during inference. However, these do not inherently accelerate the training phase.

**Novel View Synthesis from a Single Image.** Over recent years, the challenge of deriving 3D worlds from a singular image has gained traction (Rombach et al., 2021; Ren & Wang, 2022; Wiles et al., 2020; Rockwell et al., 2021). Traditional methods tackled this problem by first employing a monocular depth estimator (Ranftl et al., 2020)) to predict the 3D geometry and then utilized multi-plane images (MPI (Shih et al., 2020; Jampani et al., 2021)) or point clouds Mu et al. (2022) to craft artistic renderings. Furthermore, any gaps or inconsistencies seen from new viewpoints were rectified using a previously trained advanced neural network Yu et al. (2019). Yet, despite their efficacy, these methodologies have shortcomings. The depth estimator can be unstable, and the rectification techniques might introduce flaws that detract from the realism of the images. Follow-up approaches (Yu et al., 2021; Lin et al., 2022) propose to learn 3D priors from the 3D objects dataset and directly predict a 3D representation from a single input during the inference time. However, these methods face significant quality degradation when processing real-world images, attributable to domain differences. Xu et al. (2022) proposes to generate novel views by training a neural radiance field that only takes a single RGB-D image. Nevertheless, it needs high-quality depth as the input and only renders new views from a small range of angles. Otherworks (Xu et al., 2023; Tang et al., 2023) get rid of depth input by adopting a guided diffusion loss from a pre-trained text-

to-image model. Although these methods can render high-fidelity novel views from 360 degrees, they generally require per-scene training, which takes more than several hours.

More recently, Watson et al. (2022) propose to solve single-image novel view synthesis by using a conditional generative model. They train a diffusion model as an image-to-image translator, using the current view as the input and predicting a novel view from another angle. To extend this pipeline to in-the-wild images, Liu et al. (2023b) choose to borrow the prior knowledge from a 2D text-to-image model Rombach et al. (2022). Several other works (Liu et al., 2023a; Shi et al., 2023; Liu et al., 2023c) also consider multi-view information to enable better 3D consistency. Our proposed method mainly follows this line of works (Watson et al., 2022; Liu et al., 2023b), but significantly reduces its training cost without performance degradation.

## 3 METHOD

### 3.1 PRELIMINARIES

**Diffusion Models.** Denoising Diffusion Probability Models, simply called diffusion models, are a class of generative models that learn to convert unstructured noise to real samples. It produces images by progressively reducing noise from Gaussian noise $p(\mathbf{x}_T) = \mathcal{N}(\mathbf{0}, \mathbf{I})$, reshaping it to match the target data distribution. The forward diffusion step, represented by $q(\boldsymbol{x}_t|\boldsymbol{x}_{t-1})$, introduces Gaussian noise to the image $\boldsymbol{x}_t$. The marginal distribution can be written as: $q(\boldsymbol{x}_t \mid \boldsymbol{x_0}) = \mathcal{N}(\alpha_t \boldsymbol{x_0}, \sigma_t^2 \boldsymbol{I})$, where $\alpha_t$ and $\sigma_t$ are designed to converge to $\mathcal{N}(\mathbf{0}, \boldsymbol{I})$ when $t$ is at the end of the forward process (Kingma et al., 2021; Song et al., 2020b). In the reverse process $p(\boldsymbol{x}_{t-1}|\boldsymbol{x}_t)$, diffusion models are designed as noise estimators $\boldsymbol{\epsilon}_\theta(\boldsymbol{x}_t, t)$ taking noisy images as input and estimating the noise. Training them revolves around optimizing the weighted evidence lower bound (ELBO) (Ho et al., 2020; Kingma et al., 2021):

$$\mathbb{E}\left[w(t) \left\| \boldsymbol{\epsilon}_\theta\left(\alpha_t \boldsymbol{x}_0 + \sigma_t \boldsymbol{\epsilon}; t\right) - \boldsymbol{\epsilon} \right\|_2^2\right],\tag{1}$$

where $\boldsymbol{\epsilon}$ is drawn from $\mathcal{N}(\mathbf{0}, \mathbf{I})$, the timestep $t$ follows an uniform sampling $\mathcal{U}(\mathbf{1}, \mathbf{1000})$, and $w(t)$ serves as a weighting function with $w(t) = 1$ showing impressive results. In the inference stage, one can opt for either a stochastic (Ho et al., 2020) or a deterministic approach (Song et al., 2020a). By selecting $\boldsymbol{x}_T$ from $\mathcal{N}(\mathbf{0}, \mathbf{I})$, one can systematically lower the noise level, culminating in a high-quality image after iterative refinement.

**Diffusion Model as Novel-view Synthesizer.** The standard diffusion model serves as a noise-to-image generator, yet a modified one can be extended to an image-to-image translator, using a conditional image as the reference. By adopting a pair of posed images $\{x_0, \widehat{x_0}\} \in \mathbb{R}^{H \times W \times 3}$ from the same scene for training, an image-to-image diffusion model can take the image, $\widehat{x_0}$, as the input conditioning to predict the image from a different view, $x_0$, approximating a single-image novel-view synthesizer (Watson et al., 2022; Liu et al., 2023b). Specifically, the training objective of diffusion models becomes:

$$\mathbb{E}\left[w(t) \left\| \boldsymbol{\epsilon}_\theta\left(\alpha_t \boldsymbol{x}_0 + \sigma_t \boldsymbol{\epsilon}; t; C\left(\widehat{x_0}, R, T\right)\right) - \boldsymbol{\epsilon} \right\|_2^2\right],\tag{2}$$

where $C$ is a feature extractor, $\{R, T\}$ are the relative rotation and translation between $\widehat{x_0}$ and $x_0$. Since the task of single-image novel view synthesis is severely under-constrained, the training requires a huge dataset consisting of diverse real 3D objects. The current largest open-sourced 3D dataset Objaverse (Deitke et al., 2023) only contains 800k synthetic objects, largely behind the 5 billion in-the-wild yet annotated 2D image dataset (Schuhmann et al., 2022), not to mention the domain gap between synthetic and real samples. Liu et al. (2023b) proposes Zero 1-to-3 by initializing from a pre-trained 2D text-to-image diffusion model, successfully generalizing to real scenarios with in-the-wild images. Although Zero 1-to-3 diminishes the need for 3D training data and reduces the training cost as well, it still requires a laborious effort to converge. For instance, the official training scheme requires 10 days to run on 8 Nvidia-A100 GPUs. Our proposed method is built on top of the framework but further reduces its training time to a manageable scale.

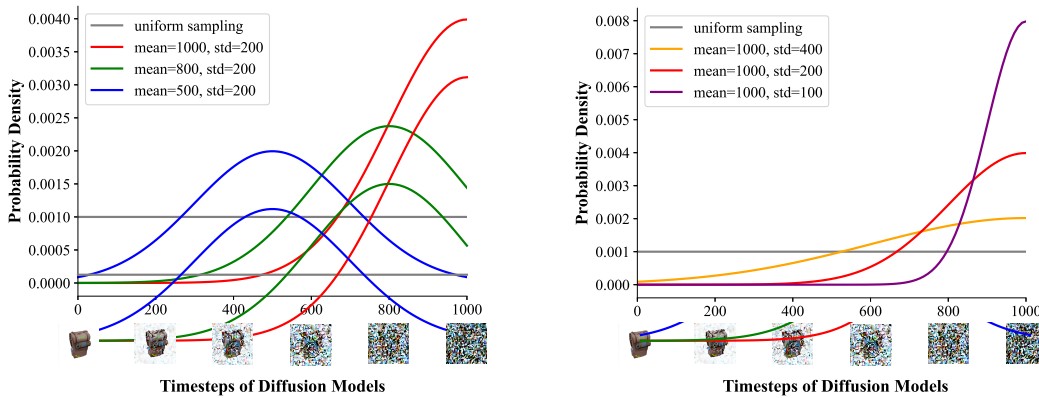

Figure 2: Gaussian sampling under different $mean$ and $std$ values. The left figure shows how the $mean$ factor could control the sampling bias towards different stages of timestep $t$, and the right figure shows how much the bias could be applied via controlling the value of $std$. The distribution is adjusted to ensure the integral of the probability density function ranges from $[0, 1000]$ is 1.

## 3.2 MODIFIED SAMPLING STRATEGIES

We start by analyzing the inference stage of the current state-of-the-art novel-view synthesizer (Liu et al., 2023b). First, we adopt denoising diffusion implicit models (Song et al., 2020a):

$$\boldsymbol{x}_{t-1} = \sqrt{\alpha_{t-1}} \underbrace{\left( \frac{\boldsymbol{x}_t - \sqrt{1 - \alpha_t} \cdot \boldsymbol{F}}{\sqrt{\alpha_t}} \right)}_{\text{`` predicted } \boldsymbol{x}_0 \text{''}} + \sqrt{1 - \alpha_{t-1} - \sigma_t^2} \cdot \boldsymbol{F}) + \sigma_t \epsilon_t \tag{3}$$

$$\boldsymbol{F} = \boldsymbol{\epsilon}_\theta(\boldsymbol{x}_t; t; C\left(\widehat{\boldsymbol{x}_0}, R, T\right)), \tag{4}$$

where $\epsilon_t \sim \mathcal{N}(\mathbf{0}, \boldsymbol{I})$ is the standard Gaussian noise, and we let $\alpha_0$ to be 1. Different choices of $\sigma$ values result in different generative processes, under the same model $\epsilon_\theta$. When $\sigma_t = 0$, the generative process degrades to a deterministic procedure and the resultant framework becomes an implicit probabilistic model (Mohamed & Lakshminarayanan, 2016). Here we do not pay attention to the stochasticity of the inference process. Instead, we analyze the output of $\epsilon_\theta$ — the "predicted $\boldsymbol{x}_0$", to better understand the learned 3D knowledge.

From Figure 1, we observe that the structure and the geometry of the predicted "backpack" are constructed in the early stage of the reverse process, while the color and texture are refined in the late stage. Note that the adopted novel-view synthesizer is initiated from a pre-trained text-to-image diffusion model, the knowledge of refining texture details should be also inherited. This observation prompts us to reconsider the prevailing uniform sampling methodology used for the diffusion timestep in the training stage. Instead, we advocate for the implementation of a meticulously designed Gaussian sampling strategy.

It is worth mentioning that the potential usage of different sampling strategies has been explored before (Chen, 2023; Karras et al., 2022). However, our work distinguishes itself, since the major training phase of our framework is essentially characterized as a finetuning paradigm. This stands in contrast to the predominant works which generally focus on training diffusion models from scratch. When juxtaposed with the uniform sampling approach, Gaussian sampling offers a distinct advantage: it permits the introduction of a sampling bias, thus conserving efforts that might otherwise be expended on superfluous segments of the training. We show several typical Gaussian sampling strategies with different mean and std in Figure 2. By adopting such a different sampling strategy, we are able to adjust the sample probability under different timesteps. Consequently, the corresponding timesteps that are still in development should receive augmented opportunities for updates, while the other parts get a lower probability, but not zero probability. From the distribution visualization, some of these Gaussian distributions satisfy our goal, such as $\{mean = 1000 \text{ and } std = 200\}$ as depicted in Figure 2. Adopting a slightly different hyperparameter may also reach a similar goal, where we simply choose an effective one by empirical observation. We show a detailed experiment to demonstrate its effectiveness in Section 4.1.1

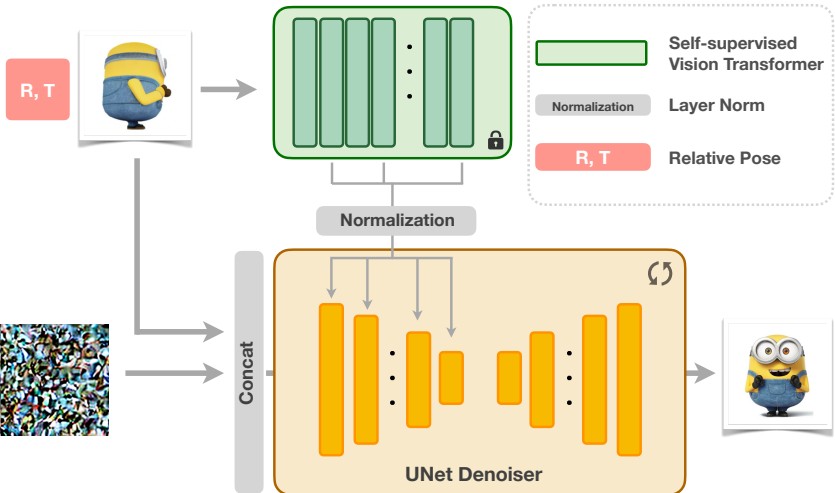

Figure 3: **Main inference pipeline of Efficient-3DiM framework.** We finetune the Stable Diffusion (Rombach et al., 2022)) on the Objaverse dataset (Deitke et al., 2023) but replace the original CLIP encoder with a self-supervised vision transformer (DINO-v2 (Oquab et al., 2023)). Rather than only adopting the "[CLS]" token from the the output of reference feature extractor, we amalgamate multi-scale representations into different stages of the UNet denoiser.

### 3.3    3D REPRESENTATION EXTRACTION AND AMALGAMATION

The task of predicting novel views from a single input view is highly ill-posed, since a singular 2D image could be projected from totally different 3D representations. Consequently, the optimization of diffusion models is expected to ensure that the 2D reference could be back-projected to the latent 3D space that aligned with human perception. Predominant works either directly concatenate the 2D condition with original noisy input and train the denoiser to capture the 3D geometry (Watson et al., 2022), or integrate an auxiliary CLIP image encoder (Radford et al., 2021) to capture the corresponding information (Liu et al., 2023b; Shi et al., 2023; Liu et al., 2023c).

However, it is noteworthy that the adopted pre-trained Stable Diffusion's foundational training task — text-to-image generation, exhibits a marginal association with the inference of 3D representations. Meanwhile, the CLIP encoder is primarily designed for the purpose of aligning with text embedding, showing poor capability on dense prediction, especially when compared with other image encoders instructed through self-supervised paradigms, E.g., DINO-v2 (Oquab et al., 2023). We show an apple-to-apple comparison between CLIP and DINO-v2 encoder in Figure 4 and 5. Through the Principal Component Analysis (PCA) and correspondence matching visualization, the spatial representations derived via the DINO-v2 encoder show more consistent semantic information than those extracted by the CLIP encoder.

Motivated by this reflection, we advocate for the incorporation of multi-scale representations produced by the DINO-v2 encoder, spanning from the output of its shallow layer to the deeper layer. Sequentially, these representations are concatenated and then converted to fit the target channel by a single linear layer. After that, we conduct several different spatial interpolation processing, and amalgamate the resultant representations into different stages of UNet's encoder, via an additive operation. We also use a cross-attention layer to combine the predicted "[CLS]" token with the intermediate feature of the UNet. Details are shown in Figure 3.

### 3.4    ENHANCED TRAINING PARADIGM

Integrating the above methodologies, we accelerate the training process by a large margin. Building on top of this, we continually introduce several engineering enhancements to optimize our framework, culminating in its most efficient version. To start with, we transition from the standard full-precision training scheme to a 16-bit mixed-precision scheme, which helps save about $40\%$ training time but leads to numerical error and causes instability. This is mitigated by adding another Layer Normalization (Ba et al., 2016) before sending the DINO-v2 feature to the diffusion model. Si-

Image Pair          CLIP          DINO-v2

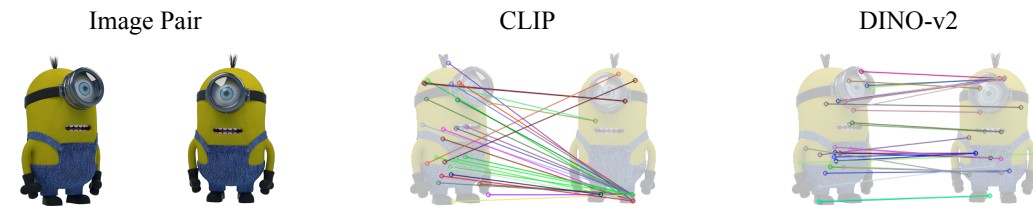

Figure 4: **Visualization of Patch Matching.** We conduct patch-level feature-matching between images under different viewpoints. Patches extracted by DINO-v2 produce better feature-matching results. More details can be found on Appendix 10.

Image Pair          CLIP          DINO-v2

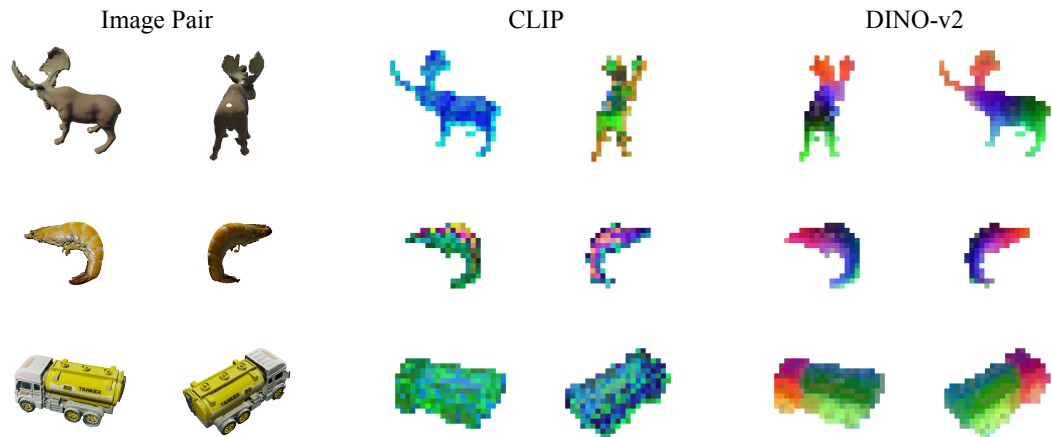

Figure 5: **Visualization of Principal Component Analysis (PCA).** For each object, we use the alpha channel to remove the white background and use PCA to condense the feature dimension to 3 on each view separately. We then visualize the features of patches by normalizing them to 0-255. Compared to the CLIP encoder, DINO-v2 exhibits a more coherent semantic understanding of identical regions, because the same parts under different viewpoints are matched regardless of poses. More details can be found on Appendix B.

multaneously, outputs from the DINO-v2 encoder are archived on the disk, given that DINO-v2 does not participate in the backpropagation chain, resulting in a $10\%$ training time reduction further. Lastly, we replace the formal constant learning rate scheduler with a cosine decayed strategy. As a result of these adjustments, we achieved a total of $50\%$ further reduction of training time, without performance degradation. Detailed training hyperparameters are included in Appendix A.

## 4 EXPERIMENTS

**Dataset** We employ the recently released Objaverse (Deitke et al., 2023) dataset for training. The dataset contains 800k three-dimensional objects, meticulously crafted by an assemblage of over 100k artists. We adopt 792k samples as the training set and 8k other samples for validation. Each object within the dataset has 12 viewpoints sampled. To make a fair comparison, we directly adopt the resultant renderings produced by Liu et al. (2023b). During the training phase, a pair of views with $256 \times 256$ resolution is sampled from each object to construct an image duo, designated as ($x$, $\hat{x}$). We randomly select one as the conditional reference and another as the ground truth view to be predicted. From this duo, the associated relative viewpoint transformation is represented as (R, T).

### 4.1 QUANTITATIVE EVALUATION

#### 4.1.1 ABLATION ON DIFFERENT SAMPLING STRATEGIES

As discussed in Section 3.2, we are motivated to switch from the standard uniform sampling of diffusion timestep $t$ to a modified sampling strategy that allows us to conserve efforts on superfluous

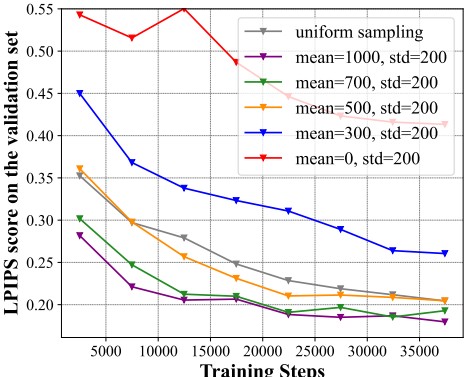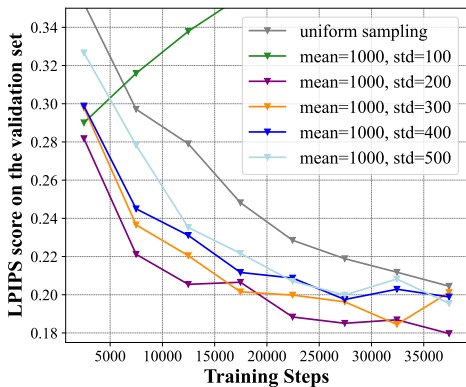

Figure 6: **Ablation study on different choices of** $mean$ **and** $std$. The left figure shows different settings by changing the $mean$ factor, and the right figure shows the results by changing the value of $std$. We report the LPIPS scores on the Objaverse validation set for all approaches.

segments of the training. Our initial assumption is built upon the observation from the diffusion model's inference process, where the noisy stage should receive augmented opportunities for updates. To verify it, we conduct experiments by controlling the value of mean and standard deviation. We first fix the standard deviation to be 200 and switch the $mean$ value from a group of $\{1000, 700, 500, 300, 0\}$, as shown in Figure 2 left. After that, we keep the $mean$ value as 1000 and test several options for standard deviation, from a group of $\{100 \sim 500\}$, as depicted in Figure 2 right. The results demonstrate our assumption — the emergence of novel view synthesis ability is more correlated with the late stage of diffusion steps (which takes a more noisy input). Meanwhile, the results also suggest that other stages should be still updated at a reduced frequency, rather than entirely omitted. For instance, in the setting of $mean = 1000$ and $std = 100$, the early stage of diffusion steps have almost zero probability of being updated, and the resultant framework shows worse results on the validation set.

### 4.1.2 Ablation Study on Different Components

We conduct an ablation study to demonstrate the effectiveness of our proposed components, as shown in Table 1. The official checkpoint of the baseline method, Zero 1-to-3 Liu et al. (2023b), requires **105,000** training iterations. Contrarily, in our experimental configuration, we constrict each counterpart to a resource-limited training scenario, allowing only **20,000** updates. Given these constraints, the baseline approach exhibits subpar performance on the validation set, only reach-

Table 1: Ablation study on different components. We train each method 20,000 steps and then report the LPIPS and MSE score on the Objaverse validation set.

| Gaussian | DINO | Amalgamation | LPIPS↓ | MSE↓ |
|:---:|:---:|:---:|:---:|:---:|
| ✗ | ✗ | ✗ | 0.236 | 0.187 |
| ✓ | ✗ | ✗ | 0.205 | 0.142 |
| ✓ | ✓ | ✗ | 0.187 | 0.136 |
| ✓ | ✗ | ✓ | 0.191 | 0.144 |
| ✓ | ✓ | ✓ | **0.171** | **0.128** |

ing 0.279 LPIPS score and 0.212 Mean Squared Error (MSE). By replacing standard uniform sampling with Gaussian sampling $\{mean = 1000, std = 200\}$, we observe improvements in both the LPIPS and MSE metrics. Next, we attempt to replace the CLIP encoder with the DINO-v2 encoder, or amalgamate the spatial feature from these encoders with the UNet denoiser. Either of them brings performance gain. Finally, we add all these modifications to the framework, reaching the best performance by obtaining a 0.171 LPIPS score and 0.128 MSE score on the validation set.

Meanwhile, we present the curve of validation loss w.r.t. the training time, shown in Figure 7. Built upon the aforementioned best setting (purple line), we further apply the proposed enhanced training paradigm (brown line). Compared to the Zero 1-to-3 baseline, the final resultant framework reaches up to **14×** speedup.

### 4.2 Visual Comparison on In-the-wild Images

We examine the performance of the obtained model on in-the-wild images. As shown in Figure 8, if we reduce the training iterations of Zero 1-to-3 from the original 105k steps to only 200k or 500k

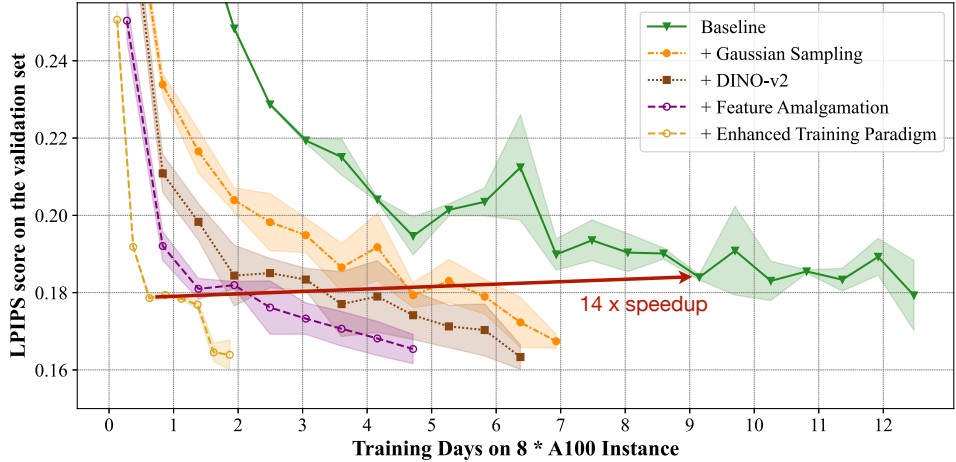

Figure 7: **Ablation study on different settings.** To verify the effectiveness of different components in our framework, we conduct three experiments for each setting with different random seeds and report the LPIPS score (Zhang et al., 2018) on the Objaverse validation set.

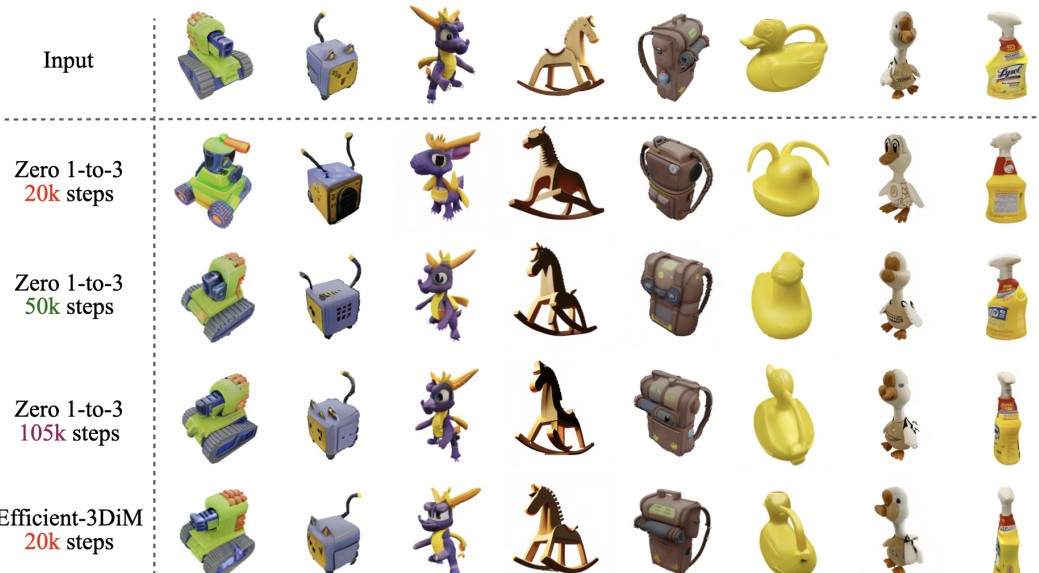

Figure 8: Visluazation comparisons on in-the-wild images. For every object, we turn the camera 90 degrees counterclockwise around the azimuth axis. More visual comparisons can be found in Appendix Figure 10.

steps, it fails to produce multi-view consistent visual outputs. However, the proposed method can generate photorealistic and geometrically reasonable novel views under the same training steps.

## 5 CONCLUSION AND FUTURE WORKS

In this paper, we introduce Efficient-3DiM, an efficient framework for single-image novel view synthesis through diffusion models. At its core, Efficient-3DiM integrates three pivotal contributions: a modified sampling strategy departing from traditional uniform sampling, an integration of a self-supervised Vision Transformer replacing the conventional CLIP encoder, and an enhanced training paradigm. Comprehensive evaluations underscore the effectiveness of proposed components, successfully slashing training time from 10 days to a single day while preserving computational resource integrity. Similar to other approaches, there remain avenues for exploration and improvement. One such avenue could involve further refinement of the multi-view consistency, ensuring both efficiency and better fidelity. Going forward, we will explore further ideas in this direction.

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

## A  DETAILED TRAINING SCHEME

All of our experiments are conducted on 8 Nvidia-A100 GPUs using the PyTorch-Lightning-1.4.2 platform. We apply a batch size of 48 per GPU and adopt gradients accumulation by 4 times. Thus the real batch size is 192 * 8 in total. We adopt an Adam optimizer with $\beta_1 = 0.9$, $\beta_2 = 0.999$, and 0.01 weight decay. We adopt a half-period cosine schedule of learning rate decaying with the base learning rate to be $1e - 4$, the final learning to be $1e - 5$, and the maximum training iterations set to be 30,000. A linear warm-up strategy is applied to the first 200 steps. We also use exponential moving average weights for the UNet denoiser. We select "ViT-L/14" for both the CLIP encoder and the DINO-v2 encoder for a fair comparison. The in-the-wild images tested on Section 4.2 are taken from (Liu et al., 2023a).

## B  PRINCIPAL COMPONENT ANALYSIS AND FEATURE-MATCHING VISUALIZATION

For Principal Component Analysis (PCA) visualization, we first use the selected encoder to capture the spatial feature maps from two viewpoints of each object. Subsequently, we employ the alpha channel to eliminate the white background. This results in $16 \times 16$ patches with $c$ channels. After that, we separately compute the Principal Component Analysis (PCA) of these two features and select the first three components. Each component is assigned a color channel, where we can further visualize it by using RGB color space. For feature-matching visualization, we directly adopt the condensed feature maps from previously conducted PCA procedures. We compute the Euclidean distance between patches extracted from two viewpoints and then map each of them. More visual results of PCA can be found in Figure 9.

## C  COMPARISONS ON REAL-WORLD BENCHMARKS

To further demonstrate the generalizability of the proposed approach, we set a testbed on an in-the-wild real-world dataset — OmniObject3D (Tong Wu, 2023) dataset. The OmniObject3D dataset contains 6,000 scanned objects with 190 daily categories. Both 2D and 3D sensors are adopted to capture each 3D object. Compared to the Objaverse (Deitke et al., 2023) dataset, the OmniObject3D

Table 2: LPIPS score on the OmniObject3D dataset (Tong Wu, 2023).

| Training Steps | Subsets of *OmniObject3D* Dataset | | | | | | | |
| | Small | | Medium | | Large | | **Average** | |
| | Zero 1-to-3 | Ours | Zero 1-to-3 | Ours | Zero 1-to-3 | Ours | Zero 1-to-3 | Ours |
| 10k | 0.326 | 0.231 | 0.326 | 0.251 | 0.330 | 0.266 | 0.327 | 0.253 |
| 20k | 0.290 | 0.224 | 0.288 | 0.246 | 0.293 | 0.274 | 0.290 | 0.248 |
| ... | ... | ... | ... | ... | ... | ... | ... | ... |
| 105k | 0.247 | - | 0.274 | - | 0.306 | - | 0.276 | - |
| 300k | 0.242 | - | 0.264 | - | 0.299 | - | 0.268 | - |

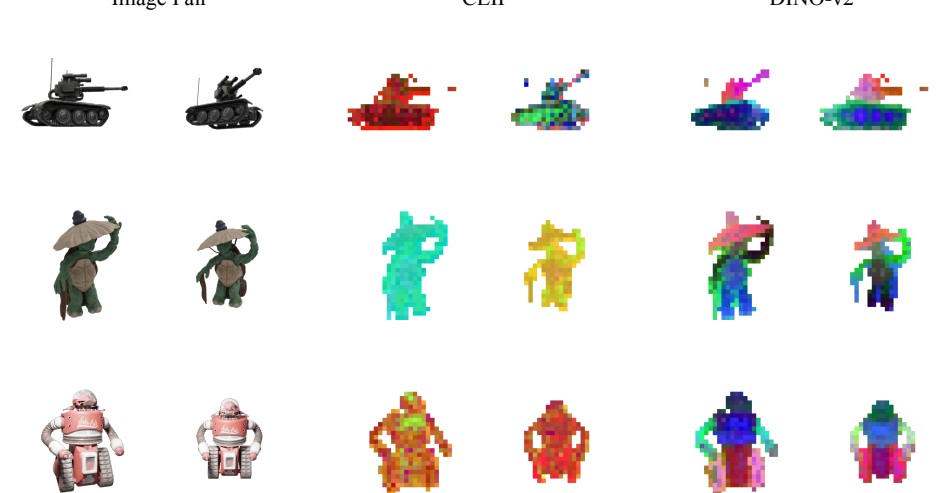

Figure 9: Visualization of Principal Component Analysis (PCA).

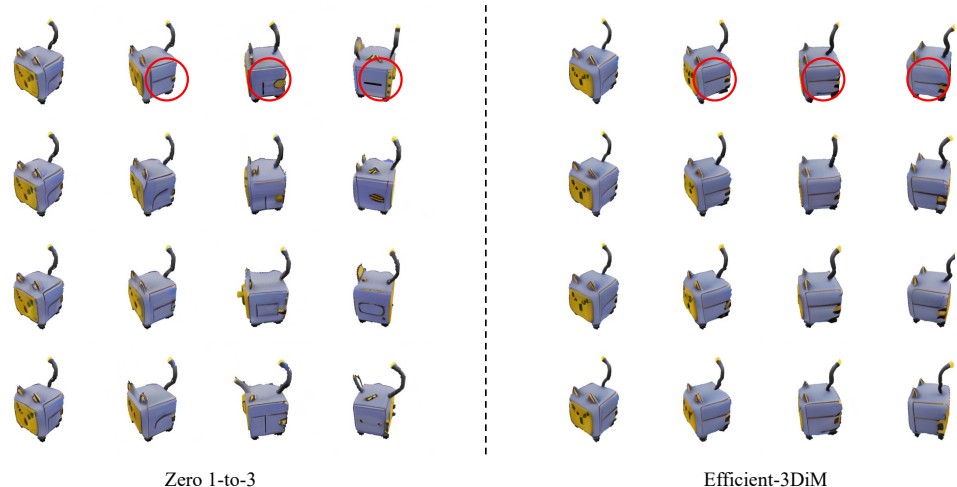

Figure 10: Visual comparisons of multi-view consistency between Zero 1-to-3 and our proposed approach. We run the inference with 4 different random seeds and different camera parameters. The highlighted region demonstrated that Efficient-3DiM shows a better multi-view consistency than Zero 1-to-3 method for every random seed we adopted.

dataset contains richer texture and more complex geometry, but less amount of data. For each object, there are more than 200 captured views. We first randomly pick a view as the canonical view and

then manually split the rest of the views into 3 categories by the rule of view distance, ranging from small (0 - 30 degrees), medium (30 - 60 degrees), and large ( $> 60$ degrees). We only sample one view from each subset and construct 4 views for each object. After that, each method is required to adopt the canonical view as the reference condition to predict the target view from these three subsets. We report the LPIPS score for each subset and also the average score. As shown in Table 2, the proposed method achieves faster training and better convergence, when compared to the baseline appraoch (Liu et al., 2023b).

## D    DETAILS OF FEATURE AMALGAMATION

To obtain multi-scale representations, we capture both the output and intermediate feature maps from the DINO-v2 encoder. The official DINO-v2 encoder contains 24 Transformer encoder blocks and the base channel is 1024. Following its original setting (Oquab et al., 2023), we select the output of the 5th, 12th, 18th, and last layers, concatenating them to a $16 \times 16 \times 4096$ feature map. It is then converted to a $16 \times 16 \times 640$ feature map through a linear layer, followed by a layer normalization layer. After that, we add this multi-scale representation to the 4th, 6th, and 8th layers of the UNet denoiser in Stable Diffusion. We only pick a subpart of this feature map depending on the number of channels in the target layer, and further conduct a bilinear interpolation on each feature map if the resolution does not match.

