# OpenReview forum: "Efficient-3Dim: Learning a Generalizable Single-image Novel-view Synthesizer in One Day"
_ICLR.cc/2024/Conference — ICLR 2024 poster_

### Official Review · Reviewer_TYx2 · 2023-10-30

**Soundness:** 4 excellent
**Presentation:** 3 good
**Contribution:** 3 good
**Rating:** 8
**Confidence:** 4

**Summary:**

This paper proposed Efficient-3DIM, a novel approach to single-image novel-view synthesis that aims to generate unseen perspectives of an object or scene from a limited set of input images. The proposed method reduces the training overhead to a manageable scale through several pragmatic strategies. Comprehensive experiments were conducted to demonstrate the efficiency and generalizability of the proposed method on common benchmarks.

**Strengths:**

Efficient-3DIM differs from previous approaches by proposing a simple yet effective efficient training framework. This is achieved through several pragmatic strategies, including a crafted timestep sampling strategy, a superior 3D feature extractor, and an enhanced training scheme.

The crafted timestep sampling strategy reduces the number of timesteps required for training, while the superior 3D feature extractor improves the quality of the learned features. The enhanced training scheme includes a self-supervised Vision Transformer and a modified sampling strategy, which are evaluated and shown to be effective in generating photorealistic novel views.

The proposed strategies significantly accelerate the training process, reducing the total training time from 10 days to less than 1 day, while shown to be effective in generating photorealistic novel views.

**Weaknesses:**

The study employs the Objaverse dataset as the sole testbed, which contains 800k three-dimensional objects. While this is a substantial dataset, its diversity and representativeness could be a limitation. I wonder if the authors would be able to demonstrate similar results on ScanNet or MVImagenet as well. Additional results on real scenes would be valuable and welcome too.

The authors mentioned that each object in the dataset underwent a procedure where 12 viewpoints are sampled. Are all comparison methods adopting the same fixed number of viewpoints?

The work itself follows the line of works by (Watson et al., 2022) and (Liu et al., 2023b), and mainly studied training efficiency improvement. The contribution is solid yet not fully substantial.

**Questions:**

Please see the weakness.

---

> ### Author Response · Authors · 2023-11-18
> **Response by Authors**
>
> We thank Reviewer TYx2 for acknowledging the novelty of our proposed framework, appreciating its effectiveness, and proposing constructive feedback. Below, we address the concerns raised in your review point by point.
>
> > Q1: The study employs the Objaverse dataset as the sole testbed.
>
> Following your suggestions, to further demonstrate the generalizability of the proposed approach, we set a testbed on an in-the-
> wild real-world dataset — the OmniObject3D dataset[a]. The OmniObject3D dataset contains 6,000 scanned objects with 190 daily categories. Both 2D and 3D sensors are adopted to capture each 3D object. We include the details of how we split the test set in the updated manuscript, as highlighted in blue color. Below we present the LPIPS score reported on the OmniObject3D dataset.
>
> | Training Steps | Zero 1-to-3  | Ours |
> | --- | --- | --- |
> | 10k.  | 0.327 | **0.253** |
> | 20k   | 0.290 | **0.248** |
> | ...   | ...  | ...  |
> | 105k | 0.276 | - |
> | 300k | 0.268 | - |
>
>
> As shown in the Table, the proposed method Efficient-3DiM converges faster than the baseline approach Zero 1-to-3. Moreover, our method only takes 20k steps to reach the best performance, better than the zero 1-to-3 method even when the latter one is being optimized by 300k training steps. More details can be found in Appendix C.
>
> > Q2: Are all comparison methods adopting the same fixed number of viewpoints?
>
> Yes, we directly use the rendered training views provided by the authors of the Zero 1-to-3 approach. For each ablation experiment, we use the same training and testing set.
>
> > Q3: The work itself follows the line of works by (Watson et al., 2022) and (Liu et al., 2023b), and mainly studied training efficiency improvement. The contribution is solid yet not fully substantial.
>
> The topic of 3D object creation has gained growing popularity in the past year. Given the importance of diffusion modes and their applications in 3D vision, developing an efficient training approach would greatly aid researchers in hastening the development process, consequently propelling the advancement of this direction. Our work is simple yet effective. The simplicity allows the proposed technique to be easily integrated into a newly designed framework in future works. We believe our work will help researchers develop novel algorithms with limited computational resources.
>
> References:
>
> [a] OmniObject3D: Large-Vocabulary 3D Object Dataset for Realistic Perception, Reconstruction and Generation

---

### Official Review · Reviewer_FP6T · 2023-10-31

**Soundness:** 4 excellent
**Presentation:** 3 good
**Contribution:** 4 excellent
**Rating:** 8
**Confidence:** 4

**Summary:**

This paper discusses the challenge of synthesizing novel views from a single image in computer vision. It introduces "Efficient-3DiM," a framework designed to significantly reduce the training time and computational costs required to train such a model, achieving a training time reduction from 10 days to less than 1 day while maintaining efficiency and generalizability through innovative strategies like non-uniform timestep sampling and improved feature extraction.

**Strengths:**

This paper was developed to enhance the training efficiency of diffusion models for single-image novel view synthesis. Core Strategies include:
- Revised Timestep Sampling: A novel strategy for selecting diffusion timesteps, and optimizing training.
- Self-Supervised Vision Transformer: Integration of a self-supervised Vision Transformer to improve the incorporation of high-level 3D features better than CLIP.
- Enhanced Training Paradigm: A refined training recipe that adopt low-precision training while addressing the numerical errors via extra layer normalization.

All strategies are grounded in motivating observations. When applied altogether, the speedup is quite significant: a 14x reduction in training time compared to the original zero 1-to-3 approach, enabling rapid iterations.

**Weaknesses:**

- Would this proposed approach be generalizable to accelerating training other image-to-3D models, such as Zero 1-to-3 and Syncdreamer? Why or why not?

- The evaluation is solely conducted using the Objaverse dataset. Although this dataset is extensive and newly introduced, relying solely on a single dataset with potential biases and limited coverage could obscure any issues that the proposed method might have in the wild. It would be beneficial if the authors could also showcase results on additional datasets for a more comprehensive assessment.

**Questions:**

Please kindly refer to the weakness section.

---

> ### Author Response · Authors · 2023-11-18
> **Response by Authors**
>
> We thank Reviewer FP6T for acknowledging the novelty of our approach to the problem and appreciating the results of our experiments. Below, we address your questions and concerns:
>
> > Q1: Would this proposed approach be generalizable to accelerating training other image-to-3D models, such as Zero 1-to-3 and Syncdreamer?
>
> Yes, our baseline method is exactly the same as the Zero 1-to-3 method, so the proposed technique is beneficial to Zero 1-to-3 as has been verified in Figure 7 of our main manuscript. The pipeline of Syncdreamer requires finetuning from a pre-trained Zero 1-to-3 checkpoint. Therefore, accelerating the training procedure of Zero 1-to-3 indeed helps reduce the total training cost of SyncDreamer approach as well.
>
> > Q2: The evaluation is solely conducted using the Objaverse dataset. It would be beneficial if the authors could also showcase results on additional datasets for a more comprehensive assessment.
>
> Following your suggestions, we set a testbed on a real-world dataset, the OmniObject3D dataset[a], to further demonstrate the generalizability of the proposed approach, The OmniObject3D dataset contains 6,000 scanned objects with 190 daily categories. Both 2D and 3D sensors are adopted to capture each 3D object. We include the details of how we split the test set in the updated manuscript, as highlighted in blue color. Below we present the LPIPS score reported on the OmniObject3D dataset.
>
> | Training Steps | Zero 1-to-3  | Ours |
> | --- | --- | --- |
> | 10k.  | 0.327 | **0.253** |
> | 20k   | 0.290 | **0.248** |
> | ...   | ...  | ...  |
> | 105k | 0.276 | - |
> | 300k | 0.268 | - |
>
>
> As shown in the Table, the proposed method Efficient-3DiM converges faster than the baseline approach Zero 1-to-3. Moreover, our method only takes 20k steps to reach the best performance, better than the zero 1-to-3 method even when the latter one is being optimized by 300k training steps. More details can be found in Appendix C.
>
> References:
>
> [a] OmniObject3D: Large-Vocabulary 3D Object Dataset for Realistic Perception, Reconstruction and Generation

---

### Official Review · Reviewer_fp2s · 2023-11-01

**Soundness:** 3 good
**Presentation:** 3 good
**Contribution:** 3 good
**Rating:** 8
**Confidence:** 5

**Summary:**

Efficient-3DIM is an efficient framework for single-image novel view synthesis through diffusion models. The proposed method reduces the training time from 10 days to a single day while generating photorealistic and geometrically reasonable novel views.

**Strengths:**

Strengths:

The authors build their work on 3DIM and integrate three pivotal contributions: a modified sampling strategy departing from traditional uniform sampling, an integration of a self-supervised Vision Transformer replacing the conventional CLIP encoder, and an enhanced training paradigm. While all those steps taken are empirical, they each carry certain contextual novelty and together yield strong training performance:
For the modified time step sampling, although similar ideas were explored before, the authors take a new angle since the major phase of 3DIM’s training is essentially characterized as a finetuning paradigm as the adopted novel-view synthesizer is initiated from a pre-trained text-to-image diffusion model.
Incorporating multi-scale representations produced by the DINO-v2 encoder, in place of the CLIP encoder, significantly improves the dense prediction and correspondence.
While the authors transit from full-precision to 16-bit mixed-precision training, they add another layer normalization before sending the DINO-v2 feature to the diffusion model, to mitigate the numerical errors.

**Weaknesses:**

Weaknesses:

(1)   I do not fully understand why LN can mitigate numerical errors, in the third part of Efficient-3DIM

(2)   The major goal of this work is to trim down the training time without spending more costs on the total training resources (e.g., taking large-batch via a distributed system). Could the authors elaborate on how the proposed method could be integrated with distributed training, and whether its speedup benefits may diminish in the (more scalable) distributed training setup?

(3)   Figure 8: I need help seeing how Zero 1-to-3 falls short of producing multi-view consistent visual outputs. All displayed results have valid visual quality to me.

(4)   Minor: “neurallift-360” paper was incorrectly cited twice in reference.

**Questions:**

Please refer to the Weaknesses

---

> ### Author Response · Authors · 2023-11-18
> **Response by Authors**
>
> > Q1: I do not fully understand why LN can mitigate numerical errors, in the third part of Efficient-3DIM
>
> The DINO-v2 encoder and UNet denoiser are separately trained under two different training tasks. Directly constructing a new model using these two components may cause instability and this issue is more severe under the mixed-precision training scheme, since 16-bit may easily introduce numerical error. The normalization layer will help calibrate the gap between these two models. We choose layer normalization rather than other types of normalization techniques due to the fact that the DINO-v2 encoder follows the design of Vision Transformer, which adopts layer normalization by default.
>
> > Q2: Could the authors elaborate on how the proposed method could be integrated with distributed training, and whether its speedup benefits may diminish in the (more scalable) distributed training setup?
>
> The proposed techniques are capable of distributed training systems and will not diminish, due to our major contribution focusing on algorithm-level only. Although we claim 192 batch size per GPU in the setting, it is achieved by 4 times gradient accumulation due to memory issues. Therefore, the real batch size per GPU is actually 48 for each single step. Under this circumstance, utilizing 4 distributed nodes and avoiding using gradient accumulation will deliver theoretically the same performance but a much faster training speed. Same as all other tasks, the speedup ratio is not ideally the same as the number of nodes we are using, due to the communication cost between each node. Improving the speedup ratio is more like a system-level task, which is currently beyond the scope of this work.
>
> > Q3: Figure 8: I need help seeing how Zero 1-to-3 falls short of producing multi-view consistent visual outputs. All displayed results have valid visual quality to me.
>
> Following your suggestions, we run each comparing method and show multiple outputs with different camera parameters as conditions. As shown in Figure 10 in the updated manuscript, we run the inference procedure under 4 different random seeds and different camera parameters. The baseline method Zero 1-to-3 fails to produce multi-view consistent textures in the highlighted region, while the proposed method successfully generates consistent results in all groups of seeds.
>
> > Q4: Minor: “neurallift-360” paper was incorrectly cited twice in reference.
>
> Thanks for pointing out the issue, we have fixed it in the updated manuscript.

---

> ### Author Response · Authors · 2023-11-22
> **Comments by Authors**
>
> Dear Reviewer fp2s,
>
> We would greatly appreciate it if you could review our response by November 22nd. After that date, it might be challenging for us to engage in further discussions. If you have any follow-up questions, please do not hesitate to reach out. We deeply value your expertise and time.
>
> Best,

---

> > ### Comment · Reviewer_fp2s · 2023-11-22
> > **About the rebuttal**
> >
> > I am happy with the rebuttal from the author, which has addressed all my concerns. Therefore, I raise my rating for this paper.

---

### Official Review · Reviewer_mro2 · 2023-11-01

**Soundness:** 3 good
**Presentation:** 3 good
**Contribution:** 2 fair
**Rating:** 5
**Confidence:** 4

**Summary:**

In this paper, the authors introduce Efficient-3DiM to accelerate diffusion models for single-image novel view synthesis, such as Zero123. Specifically, they employ a crafted timestep sampling strategy, a superior 3D feature extractor (DINO-v2), and an enhanced training scheme. Experimental results demonstrate that the proposed method could retain the performance metrics of the baseline but accomplish this with a remarkable 10x speed increase.

**Strengths:**

1. I really like the idea of integrating a self-supervised Vision Transformer for image conditions since the clip image feature only contains high-level semantic meaning. Figure 4 and Figure 5 also showcase the superiority of DINO-v2 over CLIP.
2. An in-depth and smart analysis is provided for the denoising process, and then the author proposes to sample more for larger timesteps to learn geometry, which accelerates the training process.

**Weaknesses:**

1. I am not persuaded by the motivation from comparing training image classifiers and generative models, which are not comparable. Moreover, we usually treat Zero123 as a foundation model which does not need retraining. Under this circumstance, I think the value of the proposed Efficient-3DiM diminishes.
2. The section "ENHANCED TRAINING PARADIGM" contains several well-known tricks, such as mix-precision training. I would like to suggest the authors should not emphasize this too much in their contribution.
3. I expect the proposed method with a more advanced 3D feature extractor to achieve better performance than Zero123. It is suggested to apply 3D reconstruction (e.g., Neus) on the generated views to compare Efficient-3Dim and Zero123.
4. Could you elaborate more on how to "conduct several different spatial interpolation processing"? And why do you only inject the features to the encoder of the UNet denoiser? I also suggest including more details on how to conduct feature amalgamation (e.g., feature shape and resolution).

**Questions:**

1. It is suggested to include more related work for "Novel View Synthesis from a Single Image", such as [a-d]

[a] Geometry-Free View Synthesis: Transformers and no 3D Priors. ICCV 2021.

[b] Look Outside the Room: Synthesizing A Consistent Long-Term 3D Scene Video from A Single Image. CVPR 2022.

[c] SynSin: End-to-end View Synthesis from a Single Image. CVPR 2020.

[d] PixelSynth: Generating a 3D-Consistent Experience from a Single Image. ICCV 2021.

**Details Of Ethics Concerns:**

I have no ethics concerns.

---

> ### Author Response · Authors · 2023-11-18
> **Response by Authors**
>
> We thank reviewer mro2 for appreciating our contributions and providing constructive feedback. Below we address your concerns point by point.
>
> > Q1: I am not persuaded by the motivation from comparing training image classifiers and generative models, which are not comparable. Moreover, we usually treat Zero123 as a foundation model which does not need retraining. Under this circumstance, I think the value of the proposed Efficient-3DiM diminishes.
>
> The topic of 3D object creation has gained growing popularity in the past year. Given the importance of diffusion modes and their applications in 3D vision, developing an efficient training approach would greatly aid researchers in hastening the development process, consequently propelling the advancement of this direction. In other words, reducing the training time can help researchers achieve rapid algorithm iterations. We also believe that our work enables more researchers to join this community to develop novel algorithms if they only have limited computational resources.
>
> > Q2: The section "ENHANCED TRAINING PARADIGM" contains several well-known tricks, such as mix-precision training. I would like to suggest the authors should not emphasize this too much in their contribution.
>
> Thanks for your suggestions, we have edited this part accordingly, as shown in the updated manuscript, highlighted in blue color.
>
> > Q3: I expect the proposed method with a more advanced 3D feature extractor to achieve better performance than Zero123. It is suggested to apply 3D reconstruction (e.g., Neus) on the generated views to compare Efficient-3Dim and Zero123.
>
> To verify whether a stronger 3D feature extractor can help improve 3D consistency, we conduct experiments by separately generating multiple novel views using different camera parameters. As shown in Figure 10 in the updated manuscript, we run the inference procedure under 4 different random seeds. The baseline method Zero 1-to-3 fails to produce multi-view consistent textures in the highlighted region, while the proposed method successfully generates consistent results in all groups of seeds.
>
> > Q4: Could you elaborate more on how to "conduct several different spatial interpolation processing"?
>
> The spatial resolution of the captured feature map from DINO-v2 is $(16 \times 16)$. However, the spatial resolution of the UNet denoiser varies across different stages. To match the resolution, we conduct bilinear interpolation before sending those features to the diffusion model.
>
> > Q5: And why do you only inject the features to the encoder of the UNet denoiser?
>
> We generally assume the encoder of UNet is where the 3DiM model understands the geometry of the input image and the decoder is for the purpose of reconstructing details. Therefore, adding a geometry-aware feature to the encoder may help strengthen its capability, as has been demonstrated in our experiments. This design philosophy also follows ContolNet [a], which tries to adopt add conditions to image generation. We leave the study of understanding the role of encoder and decoder for our future work, as it is currently beyond the scope of this work.
>
> > Q6: I also suggest including more details on how to conduct feature amalgamation (e.g., feature shape and resolution).
>
> Thanks for your suggestions. We include another section in the updated manuscript (Appendix D) to better describe the details of feature amalgamation module.
>
> References:
>
> [a] Adding Conditional Control to Text-to-Image Diffusion Models

---

> ### Author Response · Authors · 2023-11-22
> **Comments by Authors**
>
> Dear Reviewer mro2,
>
> We would greatly appreciate it if you could review our response by November 22nd. After that date, it might be challenging for us to engage in further discussions. If you have any follow-up questions, please don't hesitate to reach out. We deeply value your expertise and time.
>
> Best,

---

> > ### Comment · Reviewer_mro2 · 2023-11-22
> >
> > Hi,
> >
> > Thanks for your rebuttal. My concerns are addressed. I appreciate your effort in enabling more researchers to join this community. However, according to the technical part (using the DINO feature & adjusting the timestep sampling strategy), I feel like this paper is on a borderline level. Thus, I keep my rate.
> >
> > Best,
> > Review mro2

---

### Meta-Review · Area_Chair_D58w · 2023-12-05

**Metareview:**

The study introduces Efficient-3DIM, an innovative technique for synthesizing new viewpoints from a single image, streamlining the creation of unseen perspectives with a minimal number of inputs. By employing a suite of practical strategies, this approach significantly curtails training demands. The effectiveness and broad applicability of Efficient-3DIM have been validated through extensive testing across established benchmarks.

All reviewers recognize the paper's contributions to accelerating diffusion models for single-image novel view synthesis, particularly praising the integration of self-supervised vision transformers and enhanced training schemes. Despite minor concerns about the generalizability and dataset diversity, the paper's novel approach to reducing training overhead and achieving photorealistic results from limited views is seen as a solid advancement. With the authors having addressed the concerns raised, such as providing additional clarifications and expanding dataset evaluations, the consensus is that the paper deserves acceptance.

**Justification For Why Not Higher Score:**

concerns about the generalizability and dataset diversity

**Justification For Why Not Lower Score:**

The approach is simple and efficient.

---

### Decision · Program_Chairs · 2024-01-16

Accept (poster)